# Microbial community dissimilarity for source tracking with application in forensic studies

**Kyle M. Carter[1], Meng Lu[1], Qianwen Luo[2], Hongmei Jiang[3], Lingling An[1,2,4]***

**1** Interdisciplinary Program in Statistics and Data Science, The University of Arizona, Tucson, Arizona, United States, **2** Department of Biosystems Engineering, University of Arizona, Tucson, Arizona, United States of America, **3** Department of Statistics, Northwestern University, Evanston, Illinois, United States of America, **4** Department of Epidemiology and Biostatistics, The University of Arizona, Tucson, Arizona, United States of America

\* anling@email.arizona.edu

## Abstract

Microbial source-tracking is a useful tool for trace evidence analysis in Forensics. Community-wide massively parallel sequencing profiles can bypass the need for satellite microbes or marker sets, which are unreliable when handling unstable samples. We propose a novel method utilizing Aitchison distance to select important suspects/sources, and then integrate it with existing algorithms in source tracking to estimate the proportions of microbial sample coming from important suspects/sources. A series of comprehensive simulation studies show that the proposed method is capable of accurate selection and therefore improves the performance of current methods such as Bayesian SourceTracker and FEAST in the presence of noise microbial sources.

## 1. Introduction

Trace evidence, as defined by the Federal Bureau of Investigation, refers to evidence transferred by a suspect to a crime scene [1]. More commonly this is used to refer to evidence that is minute or evidence where traditional fingerprinting methods are difficult or cannot be performed, but residual information may still be collected, such as hair, soil, fiber, glass, and other environmental objects commonly found at a crime scene. Modern approaches to DNA fingerprinting involve analysis of Short Tandem Repeat (STR) and SNP gene markers that utilize PCR amplification to reduce the effect of contamination and degradation [2]. However, in trace fingerprint samples, contaminants account for a vast majority of observed DNA. Microbial alternatives to DNA fingerprinting show promise in trace evidence analysis. Direct contact can transfer millions of microbes nearly instantaneously [3]. In particular, microbial populations on items touched by hands have been found to be composed of approximately 60% to 70% of human skin-associated microbes [4]. Trace microbial profiles provide a rich avenue of exploration in tracking which suspects have had contact with trace evidence.

Microbial source-tracking refers to statistical methods which aim to identify sources of (contaminant) microbes (referred as source in this research) in an observed microbial population, i.e., sink [4, 5]. With a variety of massively parallel sequencing technologies, source-tracking methods have been employed by constructing microbial profiles using nucleotides, k-

**Data Availability Statement:** https://github.com/anlingUA/RAD.

**Funding:** This work was partially supported by the National Science Foundation [DMS-1222592 to L. A.]; and the United States Department of

Agriculture [ARZT-1360830-H22-138 and ARZT-1361620-H22-149 to L.A.]. The funders had no role in study design, data collection and analysis, decision to publish, or preparation of the manuscript.

**Competing interests:** The authors have declared that no competing interests exist.

mers, or Operational Taxonomic Unit (OTU) counts as features. By treating suspect microbiomes as potential contaminant sources and the trace evidence as a sink, microbial source-tracking methods can be applied to forensic science. Marker gene sets and REP-PCR strain-specific analyses rely on the presence of microsatellites, species that are not shared between the sources, and location of specific markers [6–8] to identify sources. However, these methods are not appropriate in forensics study. First, microsatellites are chosen based on the measured sources and are not universal; there may be sources that were not collected that contain the chosen microsatellites. If an unmeasured true source contains microsatellite features chosen from measured sources that are not the true source, these measured sources will be incorrectly selected. Second, the time difference between sample collections could be large. The temporal stability of the human microbiome may depend on the location where the samples are taken. Microbiota in human gut, nose, and throat environments have been demonstrated temporally consistent [9, 10], yet the skin microbiome may not be consistent over time and is perturbed by common activity such as hand washing or contact with others [6, 11, 12]. A study in 2014 that attempted to identify individuals based on unique microbial features saw only 13% of species-based identifiers remained after 30+ days of initial selection [11]. Community-based source tracking utilizing the full microbial profile can help dilute the effect of fluctuations [5].

In 2011, a Bayesian method SourceTracker was proposed for community-based source tracking, which estimates contamination proportions using a mixture model of OTU profiles [4]. The use of Gibbs sampling allows a probabilistic approach to calculate the mixture proportions and simultaneously accumulates dissimilar OTUs into unspecified source components. In recent years, SourceTracker has been widely used in exploratory studies involving microbial fingerprinting via surface contact with varying degrees of accuracy [4, 13, 14]. While Bayesian SourceTracker is capable of sensitivity adjustment through parameter tuning, it is not intended to test source selection and drastically increases computation times. Additionally, parameter tuning and Gibbs sampling are computationally expensive when working with large data sets.

In 2019, another source tracking method, FEAST, was proposed using an expectation-maximization with two parameter sets including mixture proportions and underlying relative abundance [15]. This approach functions similarly to the Bayesian SourceTracker except contains a much faster run time, reducing run times by a factor of 30 or more. However, similar to SourceTracker, it requires parameter tuning to achieve optimum performance and is not designed to select sources.

To improve the precision, the pool of sources can be parsed to include only the sources that contribute to the sink. In this study, we focus on developing source selection method for proportion estimation techniques using OTU profiles, which is based on the ratio of ecological similarity between sources and the sink sample. The proposed method is incorporated into existing proportion estimation algorithms and compared with Bayesian SourceTracker and FEAST using comprehensive simulation studies.

## 2. Methodology

Usually a sink sample is a mixture of two or more sources, and clustering analysis may show the correlation between them. However, traditional distance-based clustering methods will associate sink sample with sources that are similar, which may not capture true sources. Consider the relationship of a sink sample, *Ev*, and four source samples as shown in Fig 1. A majority of the sink comes from source 1, with a minor proportion from source 3 (shown in the composition of colors). Using distance-based clustering, source 2 is incorrectly clustered due to its high similarity to a true source, source 1. Meanwhile, source 3 is failed to be captured since it is much different overall from the sink. That is, the source may be wrongly attributed

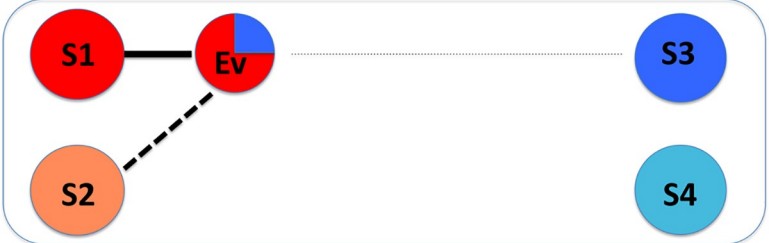

**Fig 1. Illustration of cluster based source tracking.** A visual representation of distance based clustering for sources and sink. The sink sample (Ev) is a mixture of S1 (majority) and S3 (minority). The clustering results show that it has a high relationship (solid line) with source S1 and some relationship (dashed line) with S2 and a small relationship (dotted line) with S3.

when the decision is only based on the distance to the sink (Ev). Instead, the relationship with the sink should be compared to the relationship with other sources.

## 2.1 Relative Aitchison difference source selection

Each suspect is treated as a source and labeled as $S_i$. The evidence sample is considered as a sink and labeled as $Ev$. We measure dissimilarity between two sources, represented by $D_{i,j}$ for each pair of sources $i$ and $j$. If we assume that a true source will have a higher community similarity (lower dissimilarity) to the sink than to other unrelated sources, the difference can be used to measure the association. We construct a series of hypothesis tests to represent the possible outcomes for each source $i$ given below:

$$H_0 : S_i \text{ is related to } Ev \qquad \text{vs.} \qquad H_a : S_i \text{ is not related to } Ev.$$

That is,

$$H_0 : D_{iE}^2 - D_{ij}^2 \leq 0 \qquad \text{vs.} \qquad H_a : D_{iE}^2 - D_{ij}^2 > 0,$$

for all $j$ ($j \neq i$). This test identifies whether the dissimilarity between the source and the evidence is smaller than the dissimilarity between unrelated sources. For the purpose of this study, the microbial profile for sources are formed from rarefied OTU counts data such that the total number of observed microbes is equivalent amongst all sources. Aitchison Distance is used to measure the dissimilarity. Aitchison distance preserves the unit-sum property of compositional data, making it an ideal metric to use for comparison and clustering methods for counts or proportion data [16]. Aitchison distance can be calculated using OTU counts via the following equations:

$$AD\left(X_i, X_j\right) = \left[ \sum_{k=1}^{K} \left( \log\left(\frac{x_{ik}}{g(X_i)}\right) - \log\left(\frac{x_{jk}}{g(X_j)}\right) \right)^2 \right]^{\frac{1}{2}}$$

$$g(X_i) = \left( \prod_{k=1}^{K} x_{ik} \right)^{\frac{1}{K}}$$

where $x_{ik}$ represents the abundance of $k^{\text{th}}$ OTU in source $i$. Function $g(X_i)$ refers to the geometric mean of the sample for source $i$. The difference in dissimilarity is represented as the difference between two squared Aitchison distances.

Sources that are vastly different from the others may draw a disproportionate Aitchison distance due to the presence of microsatellite features. We propose a Relative Aitchison Difference to account for unusually large Aitchison distances and ensure the measure is on the same

scale, $[-1, \infty)$, for all samples for further comparison:

$$RAD(X_i) = \frac{AD(X_i, X_E)^2 - \overline{AD(X_i, X_j)^2}}{\overline{AD(X_i, X_j)^2}},$$

where the average is taken across all other unrelated sources, *j*. With this metric, the hypothesis test can be rewritten as follows:

$$H_0 : RAD(X_i) \leq 0 \qquad \text{vs.} \quad H_a : RAD(X_i) > 0.$$

A distribution of RAD can be generated using bootstrap resampling, which resamples data with replacement. The significance of the test statistic indicates that the given source is less dissimilar to the sink than other sources and may be an important contributor to the sink sample. From here on, we will call this selection process RAD.

RAD requires the other sources for comparison to be unrelated, which may not be true in real world studies. Some examples of related sources include samples at different locations on the same subject or environment or otherwise have uncommonly similar profiles. To compensate for this possibility, Bayesian SourceTracker and FEAST use a user-defined "environment" variable to combine all samples within the same environment into a single consolidated profile. For RAD, we recommend performing hierarchical clustering to bin microbial samples. Consider once more our four sources scenario from Fig 1. Sources 3 and 4 are more similar to one-another than with the sink, but collectively they are more similar to sink than any other source. Hierarchical clustering with a single link captures these groups which may be collectively similar to the sink while avoiding drifting centroids other average linkage methods may produce, e.g., the centroid moves towards sources 1 and 2. Since our comparison will be only towards the sink sample *Ev*, bins are determined by the first nodal connection to the sink sample and all sources in the connected branch are binned together. This binning process is demonstrated in Fig 2. Additionally, the single link structure allows an easy visual representation of which branches may be collectively similar to the sink. When the sink is closer than other sources, the link distances to other sources will be shorter than link distances when the sink is removed, as represented by the red arrows in Fig 2.

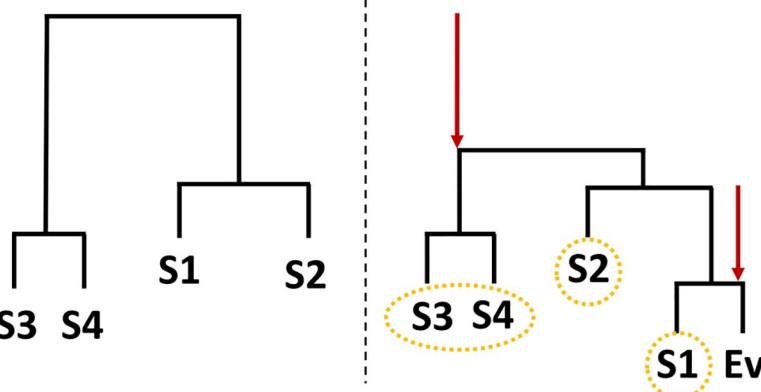

**Fig 2. Illustration of mechanism of hierarchical cluster binning.** Binning for the four sources scenario. The left dendrogram represents the connection between sources without considering the sink (Ev). The right dendrogram represents the changed relationships after the inclusion of the sink. Impacted branches are identified with an arrow, and resulting bins are circled.

When multiple sources exist within a bin, the contributor to the sink may be any or all sources within that bin. While Bayesian SourceTracker and Feast only make inference on the binned community, Relative Aitchison Difference is evaluated independently for each source within a bin, though all observations that are in the same bin as the current source are excluded when determining the average Aitchison difference. This means in the four sources scenario, we would not consider source 4 when evaluating the RAD for source 3 and vice-versa. In this example, the RAD for source 3 is calculated as follows:

$$RAD(S_3) = \frac{AD(S_3, Ev) - \left[\frac{AD(S_3,S_1)+AD(S_3,S_2)}{2}\right]}{\left[\frac{AD(S_3,S_1)+AD(S_3,S_2)}{2}\right]}.$$

## 2.2 Proportion estimation methods

**2.2.1 RAD naïve approach.**   Here we introduce a naïve approach for estimating mixture proportions based on the ratio of measured Aitchison difference introduced above. We use $I_{i_0}^*$ to indicate whether source $i$ is significant (1) or not significant (0). Proportions of microbes coming from a given source will be estimated by the ratio of mean relative Aitchison differences amongst all significant sources, as shown in the equation below:

$$P_{i_0} = \frac{I_{i_0}^* * RAD(X_i)}{\sum_i (I_i^* * (RAD(X_i)))}.$$

This method does not account for the possibility of missing sources, and will overestimate the proportions if a true source is missing from the source pool. From here on, this method is referred to as RAD-Naïve.

**2.2.2 RAD SourceTracker approach.**   RAD may also be used as a preprocessing technique for Bayesian source tracking via Gibbs sampling. By eliminating non-important sources prior to sampling, we improve the speed and predictive accuracy of this method. For comparison purposes, we have chosen to use the conditional distribution used by Bayesian SourceTracker [4]:

$$P(z_i = v|z^{-i}, x) = P(x_i|v) \times P(v|x^{-i}) = \propto \left(\frac{m_{x_i v} + \alpha}{m_v + \alpha m_v}\right) \times \left(\frac{n_v^{-i} + \beta}{n - 1 + \beta V}\right),$$

where $z_i$ is a variable representing the assigned sink (note: here we use the same mathematical notations as in the original paper). Each individual sink count is represented by $x_i$. $x^{-i}$ represents the set of all other sink counts except for the current count. Here $m_{tv}$ represents the number of sink counts from an OUT $t$ in a source $v \in \{1 \dots V\}$, while $n_v$ is the number of sink counts assigned to the source across all OTUs. Parameters $\alpha$ and $\beta$ are imaginary prior counts to smooth the distribution, and may be tuned to change sampling sensitivity. The application of Bayesian source tracking with RAD source reduction is referred to as RAD-ST.

**2.2.3 RAD FEAST approach.**   Similarly, RAD may also be used to limit the source pool for the FEAST expectation-maximization algorithm [15]. The underlying model functions similarly to the Bayesian model used by Bayesian SourceTracker, where the results are optimized by the product of relative abundance and mixture proportion. In this approach, the relative abundance for the sink is modeled:

$$\beta_j = \sum_{i=1}^{K+1} \alpha_i \gamma_{ij}$$

where $\alpha_i$ is the mixture proportion for source $i \in \{1 \dots K\}$ ($K$ is the number of known sources) and $\gamma_{ij}$ is the relative abundance of each source for each OTU $j \in \{1 \dots N\}$. The parameters are updated using the EM algorithm proposed by Shenhav et al. [5]. The application of expectation-maximization with RAD source reduction is referred to as RAD-FEAST.

**Table 1. Settings of evidence mixtures.**

| Trial | Three-source mixture | | |
|:---:|:---:|:---:|:---:|
| | Source B | Source D | Source G |
| **1** | 60% | 30% | 10% |
| **2** | 50% | 30% | 20% |
| **3** | 40% | 30% | 30% |
| **4** | 33% | 33% | 33% |

## 3. Simulation studies

Here all three methods are applied to OTU count profiles generated by 16S metagenomics sequencing. OTU data from a longitudinal study examining behavior of resident and household surface microbial communities by Lax et al. [13] was utilized. This OTU data was previously rarefied to 2500 counts per sample. Ten palm samples from the first time point were selected. These ten samples include four pairs of individuals who share a household and two individuals who are the sole household member. Each palm sample will represent a source for the sink. Our study will focus on scenarios where the sink is generated from a mixture of two to four sources. The remaining non-significant sources will be included in the source set. The sink was generated using a number of 'true' mixture models with mixture proportions as provided in Table 1 using multinomial parametric resampling. Though other combinations of sources have been examined, the mixtures mentioned in Table 1 will be used for demonstration. The RAD-naive, RAD-ST, and RAD-FEAST will be compared to Bayesian SourceTracker using 100 resamples as well as FEAST with 1000 expectation-maximization iterations. The default hyperparameters for Bayesian SourceTracker and FEAST were used, and no binning was performed for RAD because we intend to perform the selection for individuals instead of families. Furthermore, RAD methods are performed at a 5% family wise error rate adjustment using Bonferroni approach since the number of sources is relatively small. Additionally to investigate the consistence of the proposed method, we repeated our experiment using sample data from day 2~5 in Lax et al. 2015 [13].

## 4. Results

The results of two and three-source mixtures are presented below. Analyses for different time points and four-source mixtures are presented in the S1 File.

### 4.1 Three-source mixtures without missing sources

Our first analysis focuses on mixtures of three sources without any missing sources. RAD source selections showed a distinct divide in p-value between sources found significant and those not. All significant p-values were at machine 0 while non-significant p-values were approximately 1. Mean and standard deviation for estimated mixture proportions were calculated by compiling the results of twenty experimental replicates. Eight error measurements were calculated and collected for each trial and replication to compare performance between the three source-tracking approaches:

1. Root Mean Square Error (RMSE): $RMSE = \sqrt{\frac{1}{n}\sum_{i=1}^{n}\left(Y_i - \hat{Y}_i\right)^2}$

2. Relative Root Mean Square Error (RRMSE): $RRMSE = \sqrt{\frac{1}{n}\sum_{i=1}^{n}\left(\frac{Y_i - \hat{Y}_i}{\max(Y_i, \hat{Y}_i)}\right)^2}$

3. Mean Difference (MD): $MD = \frac{1}{n}\sum_{i=1}^{n}\left|Y_i - \hat{Y}_i\right|$

4. Average Residual Error (AVGRE): $AVGRE = \frac{1}{n}\sum_{i=1}^{n}\frac{|Y_i-\hat{Y}_i|}{\max(Y_i,\hat{Y}_i)}$

5. Median Absolute Deviation: $MAD = \text{median}(|x-m\text{edian}(x)|)$

6. Mean Absolute Percentage Error: $MAPE = \frac{1}{n}\sum_{i=1}^{n}\left|\frac{Y_i-\hat{Y}_i}{\max(Y_i,\hat{Y}_i)}\right|$

7. Total Variation Distance (DTV): $DTV = \sum_{i=1}^{n}\frac{|Y_i-\hat{Y}_i|}{2}$

8. Maximum Residual Error (MAXRE): $MAXRE = \max_{\text{i}}\frac{|Y_i-\hat{Y}_i|}{\max(Y_i,\hat{Y}_i)}$

where $Y_i$ represents the true mixture proportion and $\hat{Y}_i$ represents the estimated mixture proportion for source $i$. RMSE, MD, MAD, and DTV are absolute measures of error, and do not depend on the size of the true proportion. RRMSE, AVGRE, MAPE, and MAXRE are relative measures of error, where the errors are scaled based on the true proportion. For sources with a true proportion 0, the denominator is set to be the estimated proportion measures.

The results of proportion estimation for three-source mixtures (i.e., without any missing source) are provided in Figs 3 & 4. It can be seen that SourceTracker gives some proportion to the non-included sources (i.e., short green bars) as well as an unknown category (although there is no missing source), which is reflected in the error metrics plots in Fig 5. The RAD-ST result is better than SourceTracker since the unimportant sources are already filtered out in the RAD selection step.

When all sources are available (i.e., without any missing source), the addition of RAD source selection does not create noticeable improvement to FEAST model in terms of absolute error metrics (i.e. RMSE, MD, MAD, and DTV). However, since FEAST always assigns some proportion to an unknown category, the relative error metrics (i.e., RRMSE, AVGRE, MAPE, and MAXRE) result in high values for FEAST while RAD-FEAST reduces the relative errors. RAD-Naïve performs robustly across all error metrics. This experiment was repeated using a mixture with two individuals from the same family with consistent performance (See S11 –S13 Figs in S1 File). Consistent conclusions are obtained from the results of two-source mixtures (see S1 –S4 Figs in S1 File) and four-source mixtures (see S5 –S7 Figs in S1 File).

As community similarity increases, source-tracking methods may struggle to pick up differences without satellite microbes. To compare how the methods perform as similarity between important sources increases, eleven source combinations were selected to span the range of Jensen-Shannon group wise divergences for all triplets (Fig 6). These divergence rates spanned between 0.37 and 0.87. Three replicates were used for each triplet. Generally, RAD-FEAST and FEAST perform comparably best in terms of absolute error while RAD-Naïve is recommended if relative error is considered.

## 4.2 Three-source mixtures with a missing source

Our second analysis focuses on mixtures of three sources, but with the second minor source removed from the source pool. RAD source selection again showed a distinct divide between sources found significant and those not. All significant p-values were at machine 0 while non-significant p-values were approximately 1. Mean and standard deviation for estimated mixture proportions were calculated by compiling the results of twenty experimental replications. As before, RMSE, MD, RRMSE, AVGRE, MAD, MAPE, DTV, and MAXRE were calculated and collected for each trial and repetition to act as a measure of performance to compare the three source-tracking approaches. Since our pool does not contain all sources, the denominator in the RAD-Naive method does not represent all associated sources and will

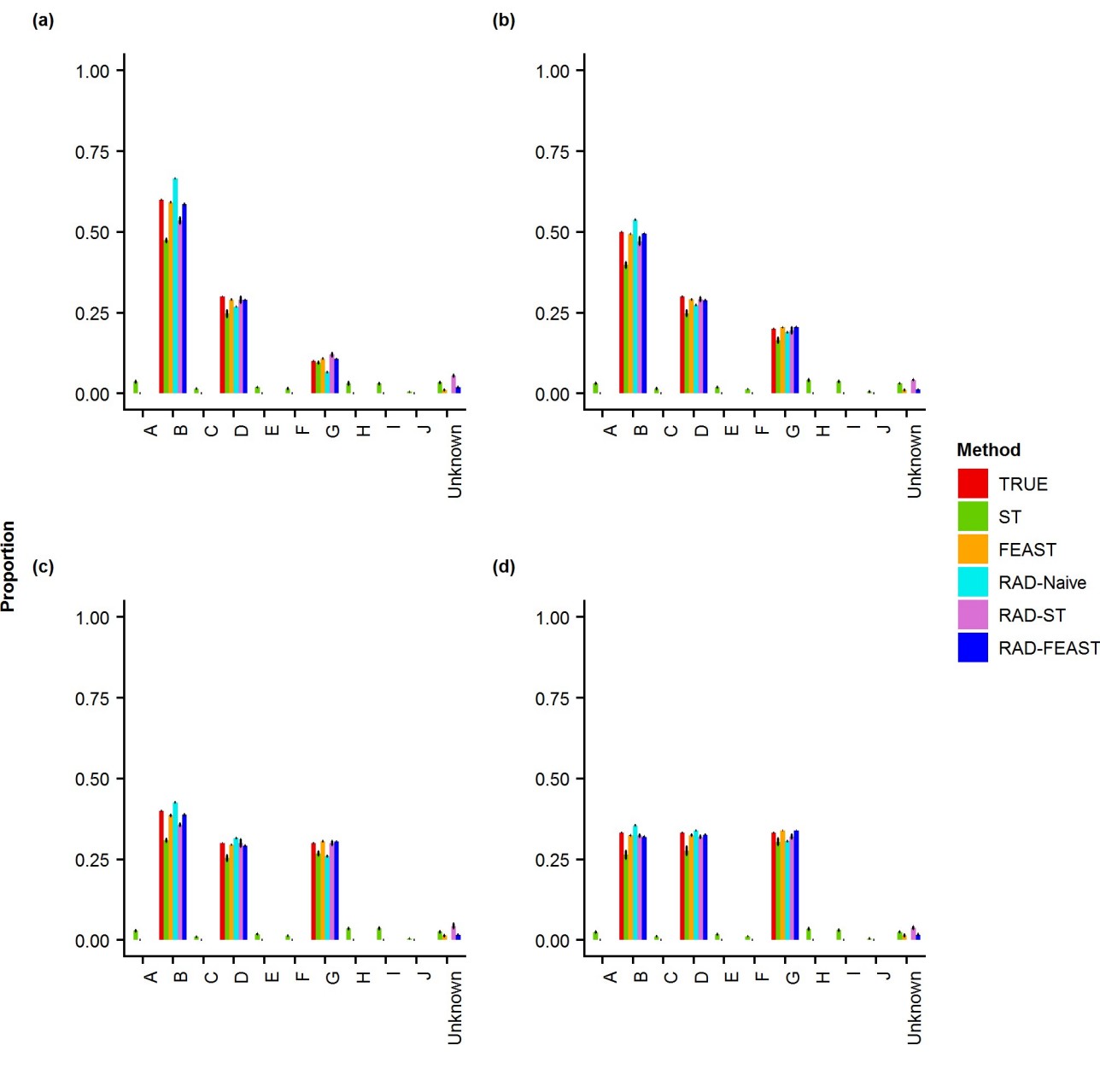

**Fig 3. Plots of proportion estimates for three-source mixture by various methods.** Comparison of true mixture proportion with estimated proportions of various mixture settings: (a) 60% - 30% - 10%, (b) 50% - 30% - 20%, (c) 40% - 30% - 30%, and (d) 33% - 33% - 33%.

overestimate proportions (Figs 7 & 8). As expected, RAD-Naive has high error rates for all mixtures (Fig 9).

When a source is missing, the Bayesian SourceTracker and FEAST both accumulate counts incorrectly to other sources, as seen with sources of I and H (Figs 7 & 8). The RAD-ST and RAD-FEAST show marked improvements over the original estimators. Particularly, RAD-FEAST performs best consistently across all settings and under various error measures (Fig 9).

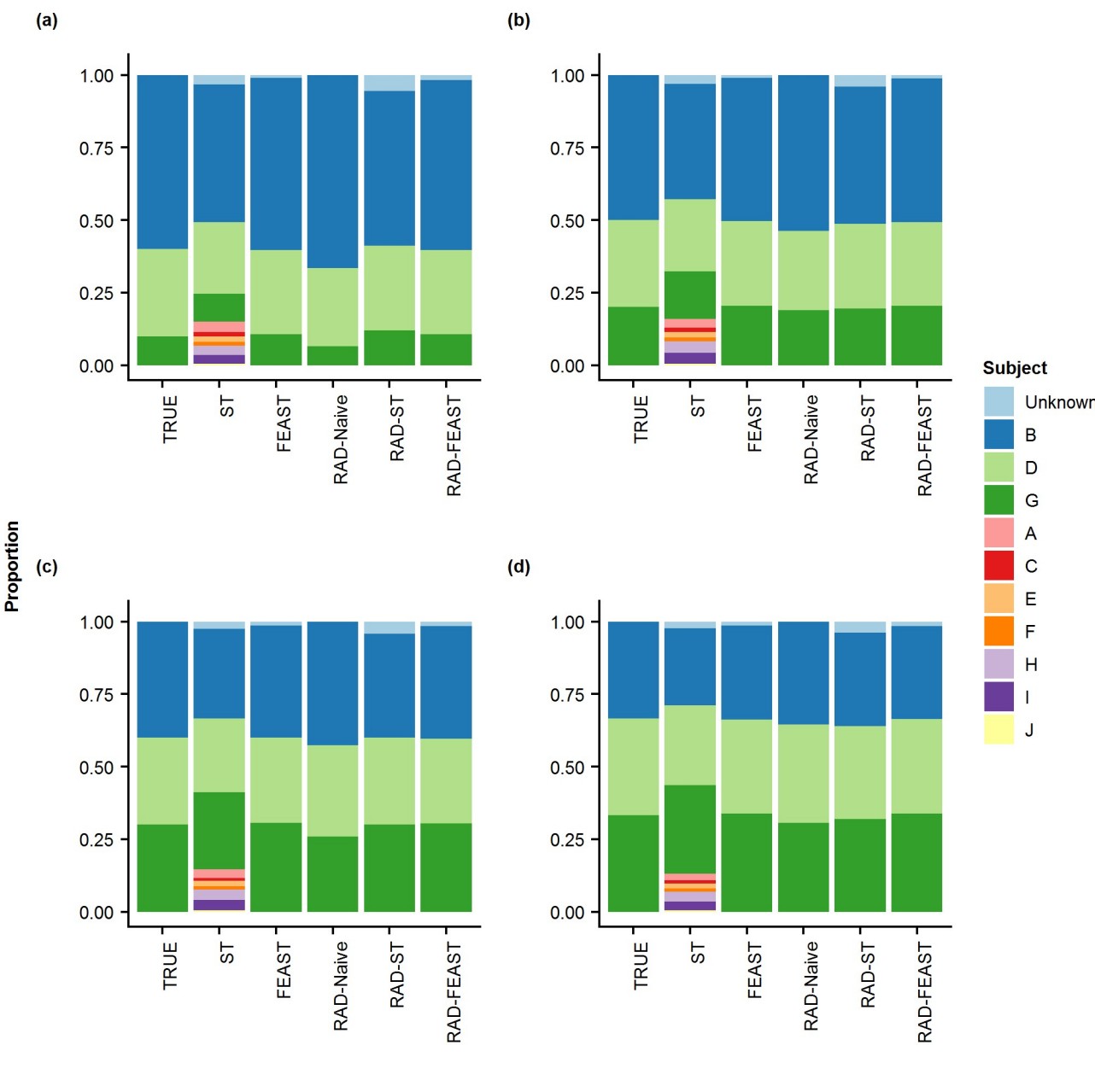

**Fig 4. Stacked-bar plots for three-source mixture by various methods.** Comparison of true mixture proportion with mean proportions of various mixture settings: (a) 60% - 30% - 10%, (b) 50% - 30% - 20%, (c) 40% - 30% - 30%, and (d) 33% - 33% - 33%.

## 4.3 Two-source mixtures with a missing source

To compare the prediction performance for various mixture proportions, evidence samples were generated using two sources across 5% increments from 0% to 100%. The two true sources and eight unrelated sources were analyzed using each method. The performance of methods was compared using RRMSE for the two true sources, i.e., not including

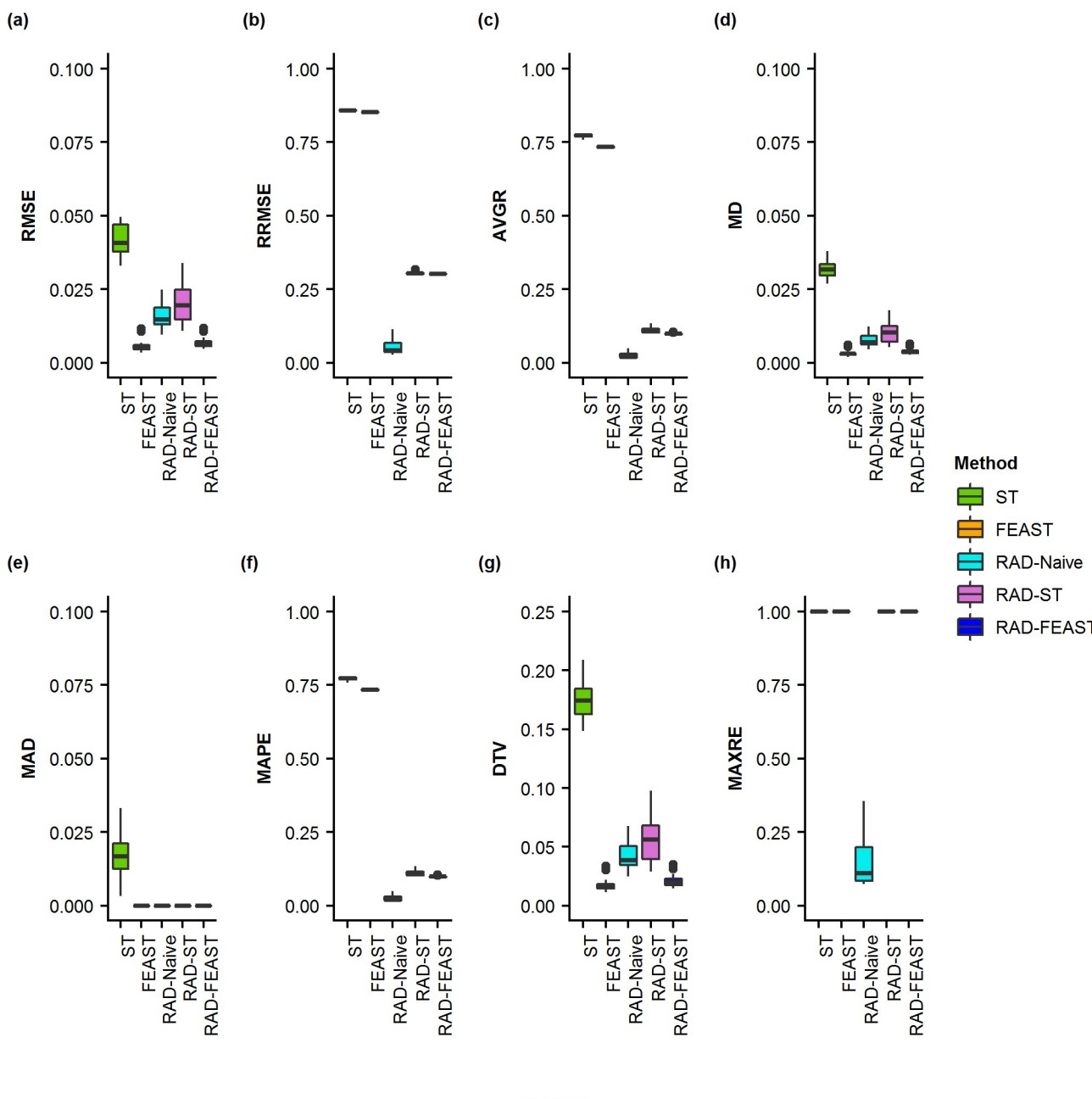

**Fig 5. Boxplots of proportion errors for three-source mixtures.** Comparison of error measurements for SourceTracker (ST), FEAST, RAD-Naive, RAD-ST, and RAD-FEAST across various mixture settings: (a) Root Mean Square Error, (b) Relative Root Mean Square Error, (c) Average Residual Error, (d) Mean Difference, (e) Median Absolute Deviation, (f) Mean Absolute Percentage Error, (g) Total Variation Distance, and (h) Maximum Residual Error.

insignificant sources. The predicted proportions are presented in Fig 10. Note that the Naïve RAD does not consider a missing source, thus the proportion of the missing source is always zero. RAD-FEAST improves the prediction accuracy for small to moderate proportions for missing sources, showing an overall decrease in the RRMSE for the two mixture components.

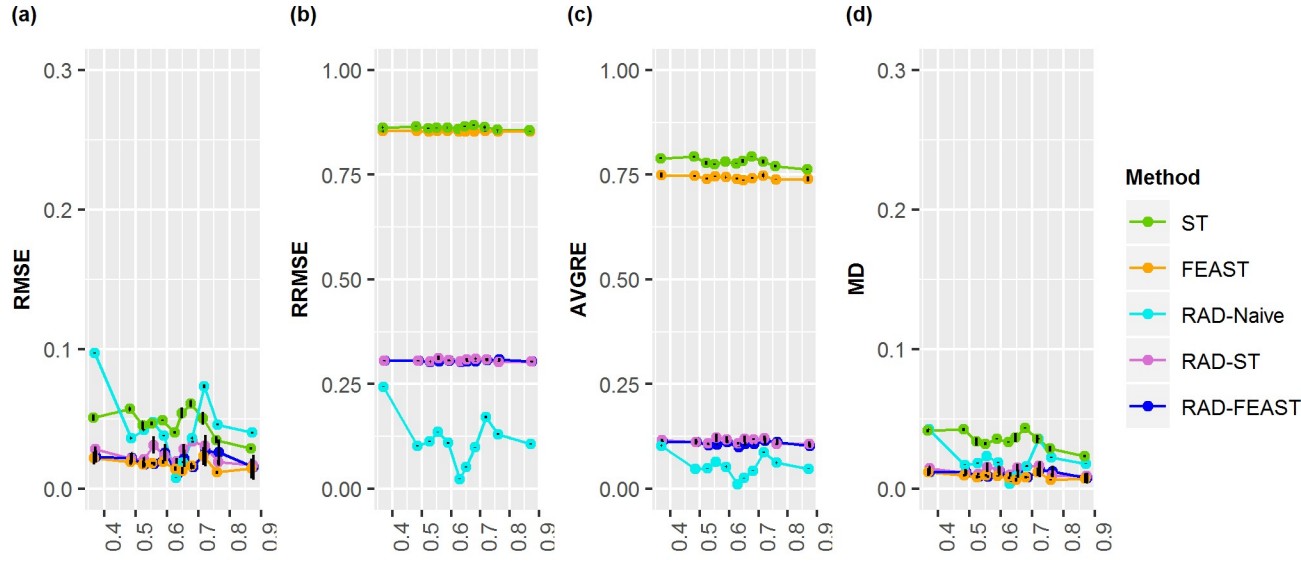

**Fig 6. Three-source proportion errors by divergence.** Comparison of error measurements for SourceTracker (ST), FEAST, RAD-Naive, RAD-ST, and RAD-FEAST for 33% - 33% - 33% evidence mixtures with varying mixture sources representing differing levels of Jensen-Shannon divergence.: (a) Root Mean Square Error, (b) Relative Root Mean Square Error, (c) Average Residual Error, and (d) Mean Difference.

## 5. Fecal source tracking application

To investigate how RAD may perform in real-world experiments, RAD was applied to an experiment by Brown et al. [17], which simulated wastewater by spiking lake water samples with bacteria from sewage effluent or fecal samples. We considered the scenario where the sources of interest are lake water, sewage effluent, cow fecal matter, and horse fecal matter. Source tracking was performed using Bayesian SourceTracker, FEAST, and RAD at genus level microbial abundances rarefied to the lowest library size of 9,000. Fig 11 displays the results for two different mixtures/evidences, one with a mixture of lake water and sewage effluent, and the other with a mixture of lake water and cow fecal matter. RAD-based methods remove all sources that are not present in the mixture sample and improve the prediction quality.

## 6. Discussion

RAD testing is able to accurately detect which sources have non-zero mixture components for each sink/evidence. In this study, the truly significant sources were selected in every replication with uncommon instances of selecting non-significant sources (S1, S2 Figs in S1 File). Integrating RAD source selection in parameter estimation has shown a sizable improvement in predicted source proportions for Bayesian SourceTracker across all settings, with both the absolute and relative error rates decreased. RAD-FEAST showed strong improvements under the missing source scenario, where unusual OTUs were attributed to another source instead of unknown.

The RAD-Naïve method performs effectively for no-missing-source cases. It tends to over-estimate larger/major proportions and under-estimate smaller/minor proportions. Due to the lack of proportion assigned to an unknown category, the relative error rates were smallest

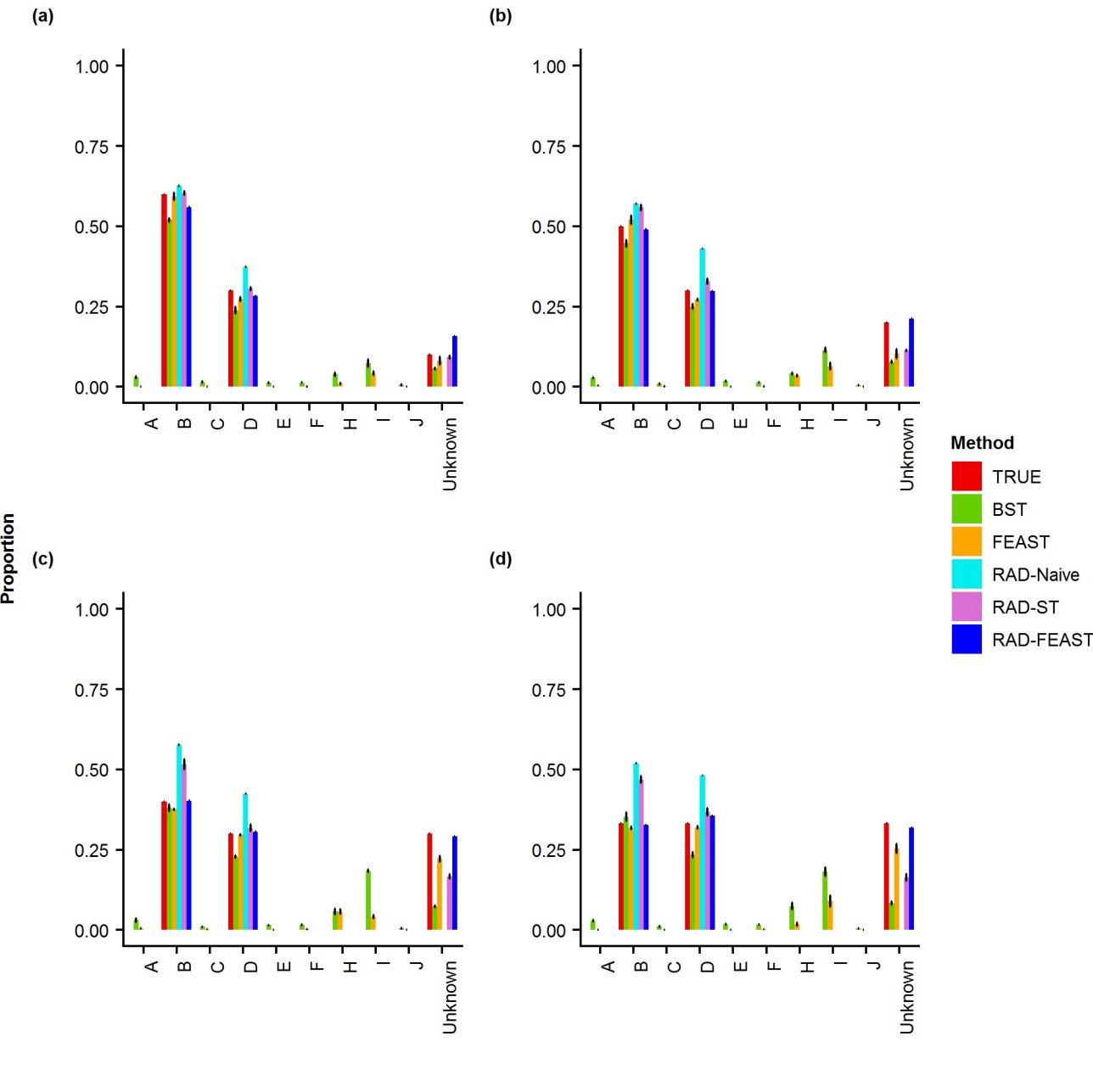

**Fig 7. Proportion estimates for three-source mixtures with a source missing.** Comparison of true mixture proportion with estimated proportions of various mixtures when the smallest component is missing from the source pool: (a) 60% - 30% - 10%, (b) 50% - 30% - 20%, (c) 40% - 30% - 30%, and (d) 33% - 33% - 33%.

when all true sources are available. Alternatively, Bayesian SourceTracker, FEAST, RAD-ST, and RAD-FEAST methods tend to under-estimate the true mixture proportions due to count accumulation into unspecified sources (i.e., the "unknown" category). In forensic practice, this feature is beneficial for its ability to take into account the possibility of missing the true culprit from our source pool.

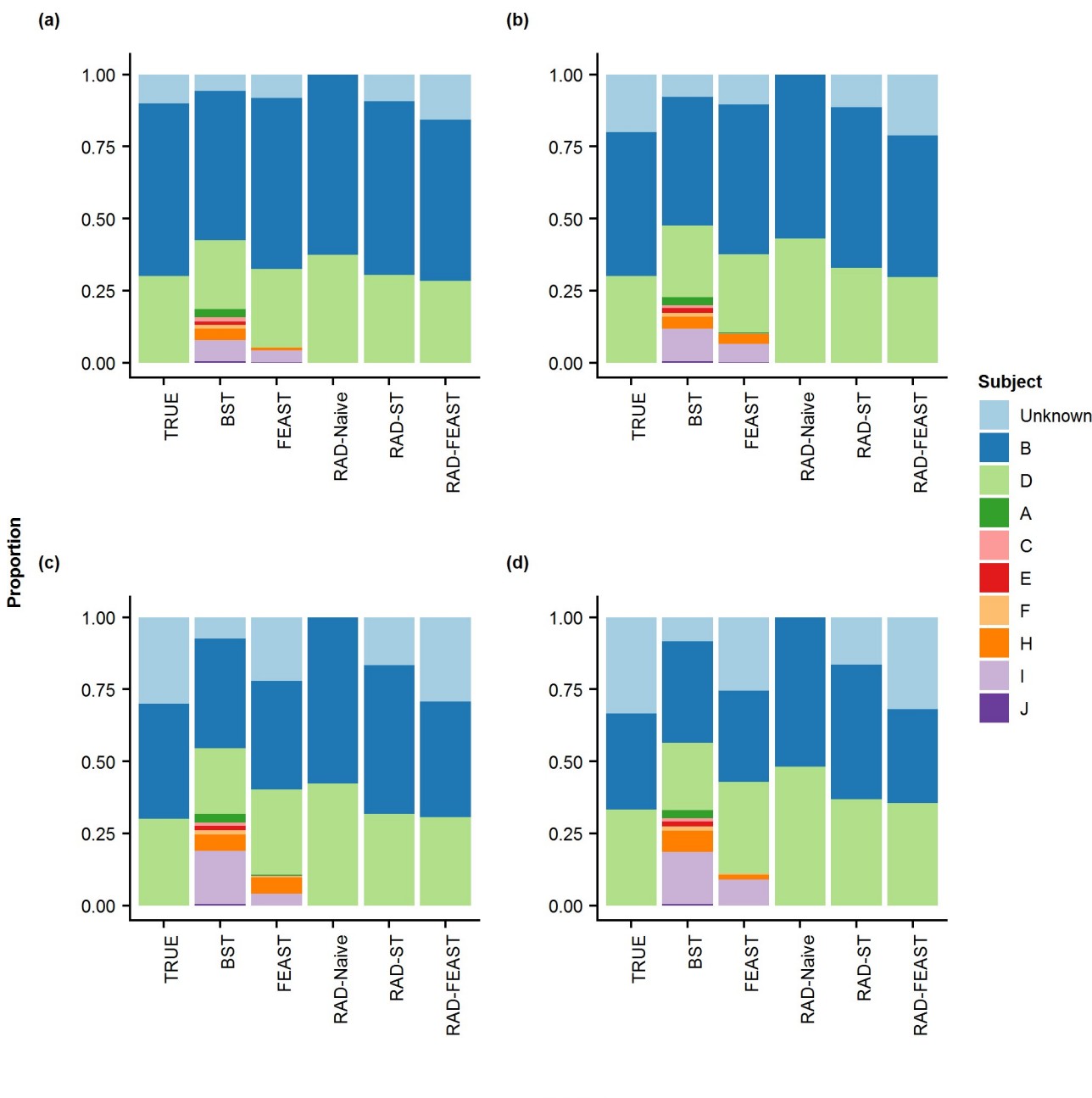

**Fig 8. Stacked-bar plots for three-source mixtures with a source missing.** Comparison of true mixture proportion with mean proportions of various mixture settings: (a) 60% - 30% - 10%, (b) 50% - 30% - 20%, (c) 40% - 30% - 30%, and (d) 33% - 33% - 33%.

Exploration across varying Jensen-Shannon divergence values revealed some trends among error rates as divergence increased. A negative relationship between performance and community divergence may be due to the limited divergence ranges covered by the ten palm samples from the original literature. Difficulties may arise when the divergence is small and within-subject random count variances overshadow between-subject variance. When applying

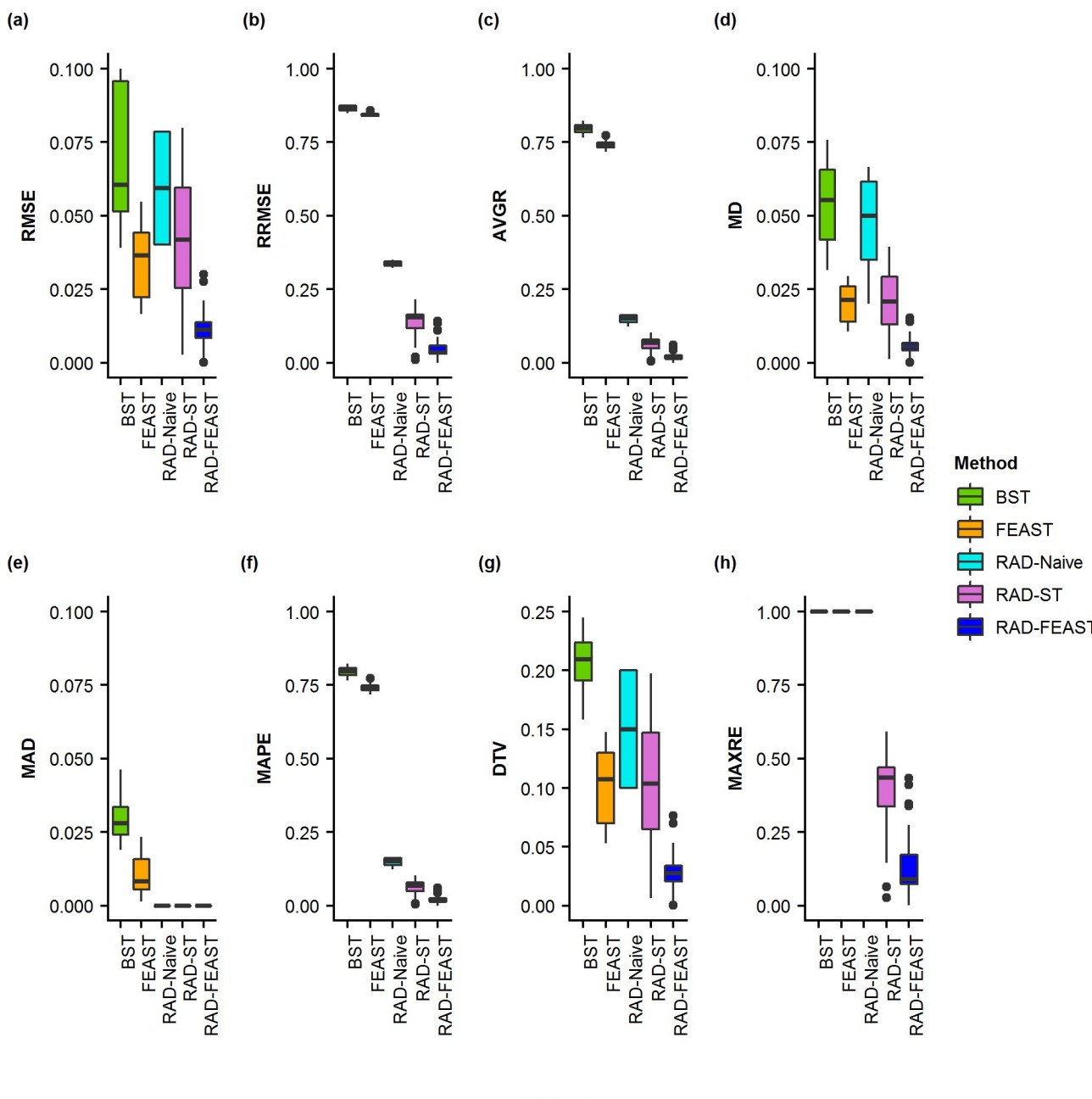

**Fig 9. Proportion errors for three-source mixtures with a source missing.** Comparison of error measurements for SourceTracker (ST), FEAST, RAD-Naive, RAD-ST, and RAD-FEAST for various mixtures when the smallest component is missing from the source pool: (a) Root Mean Square Error, (b) Relative Root Mean Square Error, (c) Average Residual Error, (d) Mean Difference, (e) Median Absolute Deviation, (f) Mean Absolute Percentage Error, (g) Total Variation Distance, and (h) Maximum Residual Error.

community-based microbial source-tracking in forensics, RAD will allow investigators to filter sources and receive estimates for proportions of microbes coming from likely sources. This technique, when combined with other forensic tools, will help investigate the likelihood that any given source is involved with the trace evidence, and by extension the crime.

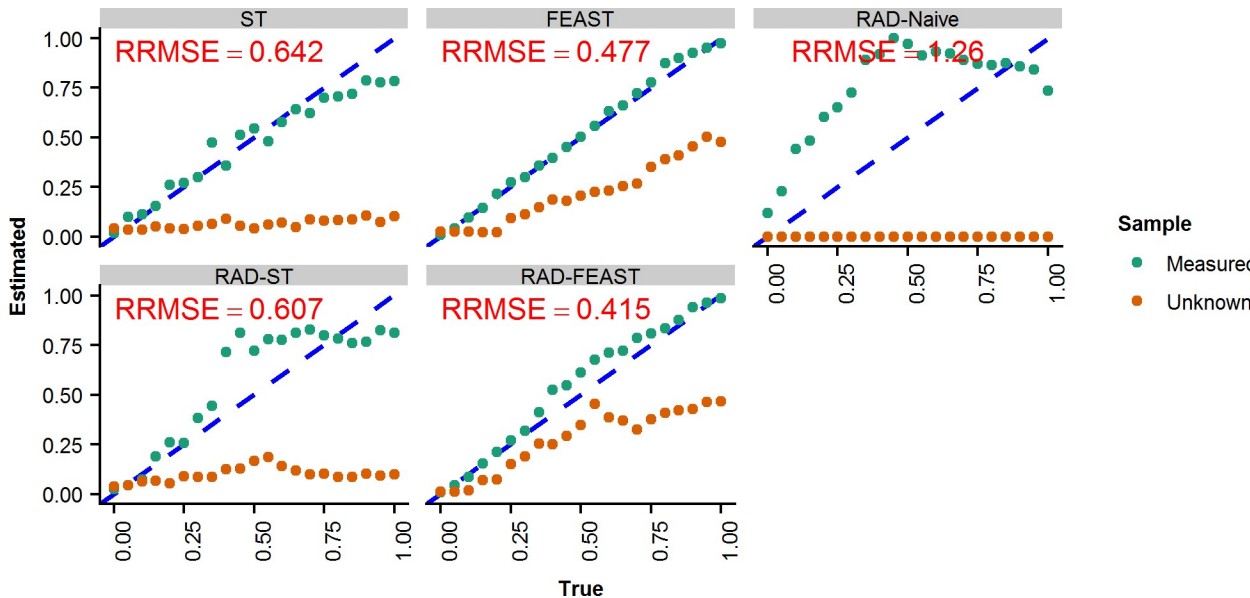

**Fig 10. Scatterplot for two-source mixtures with a source missing.** Scatterplot for true vs estimated mixture proportions for SourceTracker (ST), FEAST, RAD-Naive, RAD-ST, and RAD-FEAST from a mixture of two sources with Relative Root Mean Square Error of estimated mixture proportions of the two sources.

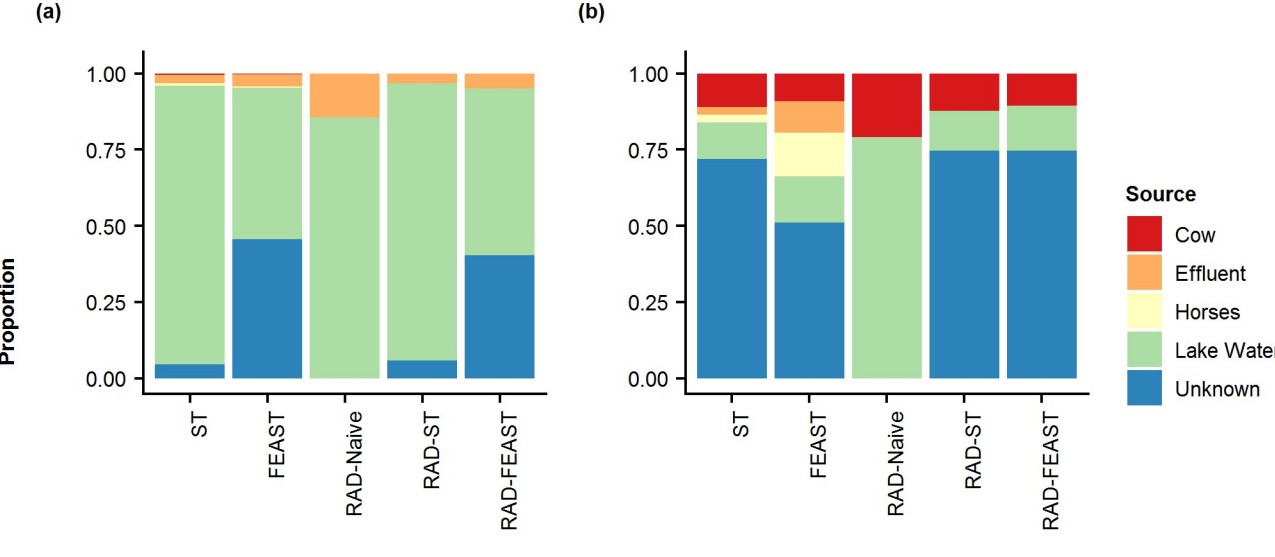

**Fig 11. Proportion estimates for spiked lake water samples.** Stacked bar plot comparing the mean proportions estimated from SourceTracker (ST), FEAST, RAD-Naive, RAD-ST, and RAD-FEAST for lake water samples spiked with bacteria from another sample: (a) sewage water effluent, (b) cow fecal matter.

# Supporting information

**S1 File.**
(DOCX)

## Acknowledgments

We heartedly thank Dr. Sadowsky's lab for the fecal data.

## Author Contributions

**Conceptualization:** Kyle M. Carter, Hongmei Jiang, Lingling An.

**Data curation:** Kyle M. Carter, Meng Lu, Qianwen Luo.

**Formal analysis:** Kyle M. Carter.

**Funding acquisition:** Lingling An.

**Methodology:** Kyle M. Carter, Meng Lu, Lingling An.

**Project administration:** Lingling An.

**Software:** Kyle M. Carter.

**Supervision:** Lingling An.

**Visualization:** Hongmei Jiang.

**Writing – original draft:** Kyle M. Carter, Lingling An.

**Writing – review & editing:** Meng Lu, Qianwen Luo, Hongmei Jiang.

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
