## [Decision Letter · Decision Letter 0]

22 Nov 2019

PONE-D-19-24969

Microbial Community Dissimilarity for Source Tracking with Application in Forensic Studies

PLOS ONE

Dear Dr. An,

Thank you for submitting your manuscript to PLOS ONE. After careful consideration, we feel that it has merit but does not fully meet PLOS ONE’s publication criteria as it currently stands. Therefore, we invite you to submit a revised version of the manuscript that addresses the points raised during the review process.

Before to proceed with further consideration of your paper all the  points raised by the  reviewers(in particular R., must be addressed)

We would appreciate receiving your revised manuscript by Jan 05 2020 11:59PM. To enhance the reproducibility of your results, we recommend that if applicable you deposit your laboratory protocols in protocols.io, where a protocol can be assigned its own identifier (DOI) such that it can be cited independently in the future. For instructions see: http://journals.plos.org/plosone/s/submission-guidelines#loc-laboratory-protocols

We look forward to receiving your revised manuscript.

Kind regards,

Giurato Giorgio, PhD

Academic Editor

PLOS ONE

Journal Requirements:

3. Please do not include funding sources in the Acknowledgments or anywhere else in the manuscript file. Funding information should only be entered in the financial disclosure section of the submission system. https://journals.plos.org/plosone/s/submission-guidelines#loc-acknowledgments

Reviewers' comments:

Reviewer's Responses to Questions

**Comments to the Author**

1. Is the manuscript technically sound, and do the data support the conclusions?

Reviewer #1: Yes

Reviewer #2: Partly

2. Has the statistical analysis been performed appropriately and rigorously? 

Reviewer #1: Yes

Reviewer #2: No

3. Have the authors made all data underlying the findings in their manuscript fully available?

Reviewer #1: No

Reviewer #2: Yes

4. Is the manuscript presented in an intelligible fashion and written in standard English?

Reviewer #1: Yes

Reviewer #2: Yes

5. Review Comments to the Author

Reviewer #1: Regarding data availability: There does not appear to be a file representing the summary data used to generate the Figures publically available. There is also no source code that I could find at: https://cals.arizona.edu/~anling/software/RAD/

Reviewer #2: In this work, the authors presented a method utilizing Aitchison distance to select important suspects (sources) for estimating the proportions of microbial communities precisely. Especially, the authors proposed Relative Aitchison Difference to consider unusual large distance. The proposed method has been integrated and benchmarked with three existing methods using simulated data sets.

Based on the results, the authors claim that the proposed method improved the prediction rate of source proportions with error rates decreasing. However, the manuscript needs to be updated considerably. Unfortunately, the actual performance of the proposed method is unclear to me. Authors do not provide a rigorous accuracy assessment and comparison with existing software tools. The results are from the relies on a relatively simple dataset (see comments below).

Major issues:

1. Page 4, Lines 72-77: The authors stated as “We propose two new methods for estimating the source proportions based on the ratio of ecological similarity between sources and the sink sample. The proposed methods are evaluated and compared with Bayesian SourceTracker using comprehensive simulation studies”. ==> However, the authors proposed to use Aitchison difference to select important sources and then adapted it to three existing methods including naïve, SourceTracker, and FEAST approaches. Therefore, the authors proposed one method, then applied it into three existing methods/tools. Please clarify it and update in the manuscript.

2. The simulated data sets have very simple proportions. The study has focused on scenarios where

the sink is generated from a mixture of two to three sources. Does the proposed method work for many sources, not just two or three? Does the proposed method work for longitudinal data sets? It is possible that the reviewer missed some important point from the manuscript. The original data set from Lax et al. ([13] in the manuscript) provide longitudinal data sets. Shenhav et al. ([15] in the manuscript) also provides many sources data set. So, the authors should provide more intensive data analysis using the data sets to validate the proposed method.

3. Figures of the results only show the prediction proportions and error bars. Please consider to choose various visualization techniques for readers to be more readily interpretable of the benefit of the proposed methods.

4. The data that the authors tested is available, but the reviewer cannot find any program scripts or software tools. Open-source package will help users to test and cite your work, so the reviewer strongly suggests to make it available in public.

6. PLOS authors have the option to publish the peer review history of their article (what does this mean?). If published, this will include your full peer review and any attached files.

Reviewer #1: No

Reviewer #2: No

---

## [Author Response · Author response to Decision Letter 0]

21 May 2020

Thanks for extending the due date for the revision. We have addressed all review comments point-by-point.

---

## [Decision Letter · Decision Letter 1]

30 Jun 2020

Microbial Community Dissimilarity for Source Tracking with Application in Forensic Studies

PONE-D-19-24969R1

Dear Dr. An,

We’re pleased to inform you that your manuscript has been judged scientifically suitable for publication and will be formally accepted for publication once it meets all outstanding technical requirements.

Kind regards,

Giurato Giorgio, PhD

Academic Editor

PLOS ONE

Additional Editor Comments (optional):

Reviewers' comments:

Reviewer's Responses to Questions

**Comments to the Author**

1. If the authors have adequately addressed your comments raised in a previous round of review and you feel that this manuscript is now acceptable for publication, you may indicate that here to bypass the “Comments to the Author” section, enter your conflict of interest statement in the “Confidential to Editor” section, and submit your "Accept" recommendation.

Reviewer #2: All comments have been addressed

2. Is the manuscript technically sound, and do the data support the conclusions?

Reviewer #2: Yes

3. Has the statistical analysis been performed appropriately and rigorously? 

Reviewer #2: Yes

4. Have the authors made all data underlying the findings in their manuscript fully available?

Reviewer #2: Yes

5. Is the manuscript presented in an intelligible fashion and written in standard English?

Reviewer #2: Yes

6. Review Comments to the Author

Reviewer #2: (No Response)

7. PLOS authors have the option to publish the peer review history of their article (what does this mean?). If published, this will include your full peer review and any attached files.

Reviewer #2: No

---

## [Editor Report · Acceptance letter]

7 Jul 2020

PONE-D-19-24969R1 

Microbial Community Dissimilarity for Source Tracking with Application in Forensic Studies 

Dear Dr. An:

I'm pleased to inform you that your manuscript has been deemed suitable for publication in PLOS ONE. Congratulations! Your manuscript is now with our production department. 

Kind regards, 

on behalf of

Dr. Giurato Giorgio 

Academic Editor

PLOS ONE